# Evaluation of miRNA as Biomarkers of Emotional Valence in Pigs

**DOI:** 10.3390/ani11072054

**Published:** 2021-07-09

**Authors:** Laura Marsh, Mark R. Hutchinson, Clive McLaughlan, Stefan T. Musolino, Michelle L. Hebart, Robyn Terry, Paul J. Verma, Stefan Hiendleder, Alexandra L. Whittaker

**Affiliations:** 1School of Animal and Veterinary Sciences, The University of Adelaide, Roseworthy Campus, Roseworthy, SA 5371, Australia; mark.hutchinson@adelaide.edu.au (M.R.H.); michelle.hebart@adelaide.edu.au (M.L.H.); stefan.hiendleder@adelaide.edu.au (S.H.); 2ARC Centre of Excellence for Nanoscale BioPhotonics, Institute for Photonics and Advanced Sensing, The University of Adelaide, Adelaide, SA 5005, Australia; stefan.musolino@adelaide.edu.au; 3Discipline of Pharmacology, Adelaide Medical School, The University of Adelaide, Adelaide, SA 5005, Australia; 4Department of Primary Industries and Regions, South Australian Research and Development Institute, Roseworthy Campus, Roseworthy, SA 5371, Australia; Clive.McLaughlan@sa.gov.au (C.M.); Robyn.Terry@sa.gov.au (R.T.); Paul.Verma@sa.gov.au (P.J.V.); 5Davies Research Centre, School of Animal and Veterinary Sciences, The University of Adelaide, Roseworthy Campus, Roseworthy, SA 5371, Australia; 6Robinson Research Institute, The University of Adelaide, Adelaide, SA 5006, Australia

**Keywords:** welfare, biomarkers, positive affective state, miRNA, pigs

## Abstract

**Simple Summary:**

It is widely recognized that the assessment of animal welfare should include measures of positive emotional (affective) state. Existing behavioral and physiological indicators of a positive affective state frequently lack sensitivity, objectivity or are unsuitable in a production environment. Therefore, there is a need to develop new approaches to accurately and objectively measure a positive emotional state in animals, including novel molecular markers such a miRNA. These biomarkers must be measurable in the peripheral circulation and provide an accurate account of the physiological and molecular activity in regions of the brain associated with emotional processing. Further, such markers require validation against established behavioral and physiological indices. Here we investigated the efficacy of circulating miRNA as biomarkers of emotional state in the pig.

**Abstract:**

The ability to assess the welfare of animals is dependent on our ability to accurately determine their emotional (affective) state, with particular emphasis being placed on the identification of positive emotions. The challenge remains that current physiological and behavioral indices are either unable to distinguish between positive and negative emotional states, or they are simply not suitable for a production environment. Therefore, the development of novel measures of animal emotion is a necessity. Here we investigated the efficacy of microRNA (miRNA) in the brain and blood as biomarkers of emotional state in the pig. Female Large White × Landrace pigs (*n* = 24) were selected at weaning and trained to perform a judgment bias test (JBT), before being exposed for 5 weeks to either enriched (*n* = 12) or barren housing (*n* = 12) conditions. Pigs were tested on the JBT once prior to treatment, and immediately following treatment. MiRNA and neurotransmitters were analyzed in blood and brain tissue after euthanasia. Treatment had no effect on the outcomes of the JBT. There was also no effect of treatment on miRNA expression in blood or the brain (FDR *p* > 0.05). However, pigs exposed to enriched housing had elevated dopamine within the striatum compared to pigs in barren housing (*p* = 0.02). The results imply that either (a) miRNAs are not likely to be valid biomarkers of a positive affective state, at least under the type of conditions employed in this study, or (b) that the study design used to modify affective state was not able to create differential affective states, and therefore establish the validity of miRNA as biomarkers.

## 1. Introduction

The assessment of emotional or affective state in animals can be challenging, particularly the assessment of positive emotion since there are fewer identified behaviors or biomarkers specific to these states. Emotions have been operationally defined as “specific, intense and short-lived responses to stimuli” whilst mood refers to “longer, more ambiguous, and nonattributable affective feelings of lower intensity” [1,2], both of which can vary along two main axes, including arousal, or strength of response, and valence (direction of response, being positive or negative) [3]. Emotions are recognized as complex, multifaceted phenomena, that give rise to rapid physiological and behavioral changes which likely evolved to achieve goals related to survival, such as attainment of valuable resources/rewards and avoidance of harm/punishment [4]. Animal welfare encompasses a long-lasting state comprising the summed-up experiences of the individual [5] and can be defined in terms of affective states and their relative weighting over time [6]. Therefore, the assessment of animal welfare should include measures of animal emotion [7]. However, in order to study animal emotional state, it is first imperative to identify methods that accurately and objectively measures the emotional state of animals.

A number of physiological and behavioral indices are currently used to infer the emotional state of animals. For example, physiological indices including hypothalomo–pituitary–adrenal axis (HPA) activity, sympathetic and autonomic functioning, endocrine function, as well as behavioral parameters have been used as makers of emotional state in animals. However, although these measures can indicate emotional arousal, they are often unable to distinguish between the valence or direction of the emotion being elicited. Furthermore, these measures tend to relate to negative affect, with less focus on, and development of, indicators of positive emotional state [8]. One assessment tool recently shown to have value in this respect is the judgment bias test (JBT), which use an animal’s behavioral response as an indicator of its underlying affective state in response to an unknown stimulus. [9]. Animals first learn to discriminate between a positive stimulus, such as a high value reward, and an aversive or nonrewarding stimulus, such as no reward or punishment [10]. Once animals have learnt to discriminate between positive and aversive stimuli, they are then tested on an ambiguous stimulus, intermediate between the two learned stimuli. These tests are based on the assumption that if, under ambiguity, the animal behaves in a manner normally associated with a positive reward, that animal has an enhanced expectation of a positive outcome that, thus, implies a positive emotional state [11]. Conversely, if the animal displays behaviors consistent with an aversive outcome, that animal has reduced anticipation of a positive outcome, which implies the animal is in a negative affective state [11]. The JBP has been used successfully in a variety of species including rats [12], sheep [13], dogs [14], chickens [15], and pigs [16,17,18], but while JBPs are considered to have good validity [19], they are less suited to production environments due to the time it takes to train animals to perform the test [20]. There is therefore an urgent need to identify and validate objective physiological or molecular markers of positive affect [21,22], in order to complement or even replace existing behavioral and physiological measures [23,24]. Following validation, new technologies may be able to be developed to analyze these biomarkers rapidly on farm using relatively noninvasive sampling, thus making them applicable for production environments (i.e., sensor-based technologies in blood or saliva).

MiRNA are small, noncoding RNA molecules involved in the regulation of genes post-transcriptionally. These molecules are ubiquitous throughout the body, including the brain, and are involved in the regulation of genes, including those associated with emotional processing [22]. For example, dysregulations of specific miRNAs have been used as diagnostic tools for a number of psychological conditions including anxiety [25,26], major depressive disorder (MDD) [27], post-traumatic stress disorder (PTSD) [28], bipolar disorder [29], and schizophrenia [30]. These molecules are involved in the regulation of emotional processes, and are released into the circulation, enabling measurement in the blood, urine or saliva [31,32]. As a result, they have the potential to be biomarkers of the activity associated with emotional processing, including those neuronal systems involved in the regulation of positive emotions such as the serotonergic and dopaminergic reward pathways [22,33,34]. For example, miRNA-16 has recently been implicated in the modulation of serotonergic transmission in the mouse brain [35]. In another mouse study, specific miRNAs, including miRNA-212, were shown to regulate the motivational properties of drug addiction within the prefrontal cortex (PFC) and striatum following the self-administration of addictive drugs [36]. Nevertheless, most miRNA research conducted in humans and rodents has focused on negative physiological or disease related conditions [37], including neuropathic pain or psychological conditions that can impact emotional state. Few studies have investigated miRNA with the specific intention to identify miRNA as correlates of positive emotional state, and to our knowledge no such studies have been conducted in pigs.

To identify and validate novel measures of positive emotion in the pig, including molecular markers such as miRNA, requires an accurate assessment of different affective states in the animal as well as the implementation of a robust means to manipulate affective state in a controlled experimental setting. Husbandry practices are known to influence production outcomes and impact welfare parameters. For example, increased floor space was shown to produce healthier pigs with high immunity and increased comfort and play behavior [38]. Pigs that are socially isolated from pen mates have shown increased behaviors indicative of stress and a decrease in behaviors indicative of positive welfare such as play [39]. The provision of enrichment to animals in farmed systems is suggested to improve biological functioning, as well as increase overall wellbeing, as it allows the animal to perform rewarding and motivated species-specific behaviors [40,41]. Furthermore, the provision of enrichment to pigs has been shown to induce a positive judgment bias compared to animals housed in barren systems, suggesting pigs provided enrichment were in a more positive emotional state [16].

In this study, we investigated the efficacy of miRNA in the brain and blood as biomarkers of positive emotional state in the pig. We anticipated that husbandry practices known to result in positive welfare outcomes would lead to a more positive emotional state in the animals compared to practices known to compromise welfare outcomes. The level of brain neurotransmitters, as well as judgment bias testing, were used as corroborating measures to infer the emotional status in pigs. We hypothesized that (i) exposing pigs to enriched housing conditions would result in a more positive judgment bias, increased neurotransmitter concentration, and differential miRNA patterns in the brain and blood compared to pigs exposed to barren environments, (ii) that changes in expression of miRNA in the brain could be corroborated with changes of miRNA expression in blood, allowing peripheral miRNA response to be used as a proxy marker for positive emotional state in the pig.

## 2. Materials and Methods

### 2.1. Animals and Housing

All animal procedures were approved by the PIRSA Animal Ethics Committee (No. 01/19), and conducted in accordance with the Australian Code for the Care and Use of Animals for Scientific Purposes (NHMRC, 2013), and the Animal Welfare Act 1985 (SA). A total of 24 female Large White × Landrace pigs with an average weight of 6.4 kg (range 5.0–8.2 kg), were selected at weaning from 12 multiparous sows and housed for two weeks in groups containing 12 animals per pen (2.0 m (W) × 4.0 m (L) × 0.8 m (H)) at the Roseworthy piggery, South Australia. During this period, pigs were exposed twice daily to 15 min of positive human interaction (patting, rubbing and scratching), and given sweet treats (M&M’s^®^, Mars Wrigley, Ballarat, Vic, Australia).

At five weeks of age all pigs were moved into group pens comprising 6 animals/pen (Figure 1). The pens were (2.26 m (W) × 4.46 m (L) × 0.86 m (H)) with flooring that consisted of half concrete and half slatted floor. Each pen contained one feed hopper, 6 nipple drinkers and an overhanging heat lamp that was turned on daily between 18:00–06:00. Pigs had access to water and ad libitum grower feed (Barastoc MP Pig 1300, Ridley’s, Adelaide, SA, Australia).

### 2.2. Spatial Judgment Bias Task

From 5 weeks of age pigs were first trained to perform in a judgment bias test that consisted of a spatial, go/no go task. During the training phase pigs discriminate between positive and aversive stimuli within a test arena (Figure 2A). Each stimulus was associated with two cues, (1) bowl color (blue = positive and red = aversive) and (2) bowl location (right or left). Each cue was reinforced with either a food reward (M&M’s, positive) or no food reward plus a scare from human (see below, aversive). To ensure pigs could not discriminate between the positive and aversive reinforcer, the red bowl (aversive cue) contained chocolate treats that were unattainable to pigs due to a plastic covering (Figure 2B). The location and color of bowl were randomized for each pig using computer generated randomization in Excel (Microsoft Excel 2016, Microsoft Corporation, Redmond, WA, USA). Ordering was consistent for each pig across time. Pigs that did not learn to discriminate between the positive and aversive stimulus during training were excluded from the analysis. Exclusion criteria were based on previous literature [42], where pigs were excluded if their individual mean latency to approach the aversive cue was equal to, or lower than their individual mean latency to approach the positive cue on their last day of training. The timeline of training is provided in Figure 3 and details are provided below.

#### 2.2.1. Training Protocol

##### Week 1 Training

During week 1, pigs were habituated to the test arena once a day for two days. On each training day, pigs were exposed to ten consecutive trials (3× group for 300 s, 4 × group for 240 s and 3 × individual for 60 s). Each group trial consisted of the 3 animals housed in the same pen. Pigs entered the arena and were allowed to familiarize themselves with the arena and the positive stimulus. For each positive stimulus the positive cue was placed (no lid), at either left or right side of the testing arena and was filled with chocolate treats (M&M’s). If pigs had not approached the bowl by the end of the trial, they were given additional time to approach the positive stimulus and eat the sweet treats. If pigs in individual trials displayed distress, i.e., high pitch screams, escape attempts, erratic movements or loud grunting, the individual was removed from the test arena and an additional group run was performed thereafter. Following an additional group run the pig was then trialed individually until all trials were completed.

##### Week 2–5 Training

Animals were trained twice a week on alternate days. On each training day pigs were exposed to eight individual consecutive trials of 60 s each. During week two training, the lid remained off the bowl in the positive stimulus. The positive cue contained five sweet treats and pigs that approached the bowl were allowed to eat the treat before being removed from the arena. From week 3 of training the lid was placed on top of the bowl and remained on for the remainder of the training and testing sessions. If the pig approached the bowl and flipped the lid, it was considered a pass and the pig was allowed to eat the treat before being removed from the test arena. If the pig failed to flip the lid it was considered a fail. Training continued until all pigs passed and were able to flip the lid within 60 s upon entering the arena.

##### Week 6–10 Training

Pigs were trained individually twice a week on consecutive days where the aversive stimulus was introduced. Each day pigs performed 5 trials (individual for 60 s that comprised 3× positive and 2× aversive cues). The order of trials was pseudorandomized so that no more than two positive or aversive cues were conducted in secession, but the final trial was always positive and was adapted from similar training protocols conducted previously in pigs [43] and sheep [44]. Six trials were deemed sufficient per training session based on learning ability during training weeks 1–5. If pigs approached the aversive stimulus and flipped the bowl lid, an investigator holding a toy clapper would move the clapper vigorously close to the pig’s face until the pig retreated. The pig was then removed from the arena. Pigs who approached the positive stimulus were allowed to eat the reward before being removed from the arena.

##### Refresher Training

Refresher training occurred once a week between test 1 and test 2 (JBT1 and JBT2, respectively). This was performed to reinforce the associations between positive and aversive cues between the first and second tests. The refresher training followed the same training protocol as week 6–10 training (see above).

Once trained, pigs then underwent two judgment bias tests where the ambiguous stimulus was introduced and included a black bowl placed between the positive and negative stimulus and was unrewarded with treats. JBT1 occurred following week ten training and was prior to treatment allocation, and JBT2 occurred four weeks later following treatment allocation.

#### 2.2.2. Testing Protocol

The test protocol was the same for both JBT1 and JBT2. Each test consisted of eight consecutive trials of 60 s each, and the trial order remained the same for each pig being tested (P, N, P, A, N, P, N, A). The sequence of trials was planned to ensure that, for all animals, the number of times each ambiguous location followed immediately after a rewarded location, and immediately after an unrewarded one, was the same [17]. The test period began when pigs moved from the start box and both front legs had entered the test arena and ended after 60 s. Latency to approach bowl was recorded for each trial using a stopwatch and times were confirmed with video data derived from one video camera (HERO5, GoPro Inc., San Mateo, CA, USA) mounted on either side of the test arena. The stopwatch was started when the pigs two front legs entered the testing arena from the start box. Following the last trial each pig was moved away from the testing area and a blood sample was taken before the pig was returned to its home pen.

### 2.3. Treatments

Treatment allocation was randomized from JBT1 data so that each treatment group comprised the same number of pigs classified as having either positive bias, negative bias or unknown bias. Individual pigs who took longer to approach the ambiguous stimulus in JBT1 relative to the mean latency of all pigs to approach the ambiguous stimulus in JBT1 were considered to have negative bias. Conversely, individuals who took less time to approach the ambiguous stimulus relative to the mean latency of all pigs were considered to have positive bias. Individuals on the mean were considered unknown bias and randomly allocated between treatments. Pigs were then allocated between two treatments with *n* = 12 each: barren housing or enriched housing (Figure 4). Barren housing entailed animals being individually housed in barren stalls (0.6 m (W) × 2.24 m (L) × 1.7 m (H)), where pigs had sight of neighboring pigs but were unable to physically interact. Each stall contained a feed hopper and nipple drinker. Pigs had access to water and were fed 4 kg standardized grower feed (Barastoc MP Pig 1300, Ridley’s, Adelaide, SA, Australia) daily. No human contact was present except for the person feeding and cleaning in the morning. In enriched housing pigs were in groups of 3 per pen (2.0 m (W) × 4.0 m (L) × 0.8 m (H)) and exposed to positive human contact (patting, rubs and scratches) for 15 min daily. Toys were also provided for enrichment and included tennis balls, basket balls, chains, ropes and PVC piping, and rubber matting. Each day the toys were placed back into the appropriate pen so that each pig had access to one of each type of toy continuously. The choice of enrichment was based on previous studies investigating the effects of providing various enrichments on welfare parameters in pigs [41,45,46,47].

### 2.4. Blood MicroRNA Collection

Immediately following JBT1 and JBT2, pigs were restrained using a rope snare and a 3 mL blood sample collected from the jugular vein of each pig into a 4 mL-Lithium-Heparin coated tube (Vacuette, Greiner Labortechnik, Kremsmünster, Austria). Following this, 500 µL of whole blood was aliquoted into 1mL animal blood tube (Qiagen, Hilden, Germany). The blood tubes were then stored at 4 °C for 24 h and then frozen at −80 °C, following manufacturer guidelines, until further analysis.

### 2.5. Brain MiRNA and Neurotransmitter Collection

One day following JBT2, 6 randomly selected animals of each treatment were humanely killed with 1 mL/10kg i.v. of pentobarbital sodium (Virbac Pty Limited, Milperra, Australia) and the brain removed immediately following protocol developed by Bjarkam et al. [48]. The remaining twelve animals were returned to the commercial herd. Once removed from the skull, the brain was then submerged in ice cold saline and then sectioned into right and left cerebral hemispheres. The right cerebral hemisphere was placed directly into liquid nitrogen and frozen at −80 °C for subsequent HPLC analyses. The left cerebral hemisphere was sectioned into 5 mm coronal sections (rostral to caudal, Figure 5), and each section placed in a 150 mL specimen tub containing 100 mL of RNA stabilizing solution and then stored at −20 °C.

### 2.6. Extraction of miRNA

A stereotaxic atlas of the pig brain [49], was used to identify the amygdala. Using a 1 mm biopsy punch (Ted Pella, Redding, CA, USA), a sample was taken from the amygdala (see Figure 6), weighed and immediately underwent extraction of total RNA. Isolation of total RNA was performed from the blood and tissue samples using RNeasy protect animal blood kit (Qiagen, Hilden, Germany), and RNeasy plus Universal kit (Qiagen, Hilden, Germany), respectively, according to the manufacturer’s instructions. Integrity of RNA was determined using 2200 Tape-Station Analysis software (Agilent, Mulgrave, Australia), and samples with RIN values greater than 7.5 were used in the analysis.

### 2.7. Expression Profiling of miRNA

Differentially expressed miRNA in blood and amygdala RNA were detected using Affymetrix gene chip technology (GeneChip™ miRNA 4.0 Array, Thermofisher Scientific, Thebarton, SA, Australia), and performed by ACRF Cancer Genomics Facility (Centre for Cancer Biology, SA Pathology, Adelaide, SA, Australia), in accordance with manufacturer’s instructions. Briefly, poly(A)Tailed, biotin labelled miRNA was prepared from 500 ng of total RNA sample using the FlashTag Biotin HSR RNA Labelling Kit for GeneChip miRNA Arrays (Thermo Fisher Scientific, Thebarton, SA, Australia, cat. no. 901910). Labelled RNA samples were hybridised to GeneChip miRNA v4.0 arrays with arrays incubated in a GeneChip Hybridization Oven 645 for 16 h at 48 °C. Array washing and staining were performed on the GeneChip Fluidics Station 450, and scanned using GeneChip Scanner 3000 7G. CEL files were generated using Affymetrix GeneChip Command Console Software v4.0 (Thermo Fisher Scientific, Thebarton, SA, Australia).

### 2.8. HPLC Analysis

Regions of the brain including the striatum, amygdala and prefrontal cortex were dissected working on ice from the right cerebral hemisphere using the stereotaxic atlas of the pig brain derived from Félix et al. [49]. High performance liquid chromatography (HPLC), analysis was conducted to detect dopamine (DA), serotonin (5-HT), and their respective metabolites (DOPAC and 5-HIAA), using previously published methodology [50].

### 2.9. Statistical Analysis

#### 2.9.1. Behavior

Behavior data were analyzed in statistical software package IBS SPSS to investigate differences in judgment bias between treatment groups. All behavior data were tested for normality and homogeneity and nonparametric analysis was conducted where appropriate. Training data were analyzed using a Friedman test to determine differences in latency to approach positive and aversive cues over time (training week 1–10 for positive and training weeks 6–10 for aversive, *n* = 24). A Wilcoxon signed-rank test was then conducted to determine differences between individual weeks. A Kruskal–Wallis test was then performed to determine difference in latency between positive and aversive cues at week ten of training.

JBT1 data were analyzed using Mann–Whitney–Wilcoxon test to determine differences in latency towards cue location, and was performed on 23 pigs (*n* = 12; enriched, *n* = 11; barren), as one pig had to be euthanized on humane grounds. Kruskal–Wallis and Mann–Whitney–Wilcoxon tests were then performed to look at treatment effects on latency towards the ambiguous cue at JBT1 and JBT2 and between JBT1 and JBT2. To control for possible intrinsic differences between pigs (i.e., walking speed, food motivation and body size), an adjusted judgment bias index (JBI), was calculated for each pig at JBT1 and JBT2 following a formula described by Horback et al. [43]. The JBI normalizes the animal’s response toward the ambiguous stimulus based on its previous responses to the positive and negative stimulus. The index ranges from 0–1 where animals with a JBI < 0.2 are considered negatively biased, a score of > 0.8 are positively biased and a score between 0.3–0.7 are unknown bias. A Fisher’s Exact Test analysis was performed to determine the change in proportions in JBI between pigs exposed to positive or negative housing at JBT1 and JBT2. Latency data are presented as medians with upper and lower range and JBI data are presented as proportions. Data were considered significant when *p* ≤ 0.05 unless stated otherwise.

#### 2.9.2. Blood and Brain MiRNA

Analysis of differentially expressed genes in blood and brain were conducted following a similar statistical protocol performed previously [51]. Affymetrix data were imported into genomic software package TAC (Transcriptome analysis console 4.0, Applied biosystems, Thermofisher Scientific, Thebarton, SA, Australia). Independent *t*-tests to determine between and within treatment effects at bleed 1, bleed 2 and in Amygdala were performed. Differences were considered significant when a gene level fold change of <2 or >2 occurred with an FDR adjusted *p*-value of less than 0.05 (FDR *p* < 0.05).

#### 2.9.3. Dopamine, Serotonin and Metabolites

Brain dopamine (DA), serotonin (5HT), and their respective metabolites DOPAC and 5H1AA were analyzed in statistical software package IBS SPSS to investigate differences in expression between treatments. Data were tested for normality and homogeneity using the Kolmogorov and Levene’s test, respectively. A Mann–Whitney–Wilcoxon test was then performed to investigate treatments differences in Amygdala, Striatum and Prefrontal cortex. Data are presented as medians ± range with a significance level of *p* < 0.05.

## 3. Results

### 3.1. Behaviour Data

#### 3.1.1. Identification of Positive and Aversive Cue

During the learning phase (weeks 1–10) pigs were able to successfully identify the positive cue as shown by the decreased mean latency to approach the positive cue over time (χ^2^ (9) = 117.7, *p* = 0.000, Figure 7A). During the learning phase from weeks 6–10 there was a significant difference in the latency towards the aversive cue over time (χ^2^ (4) = 12.99, *p* = 0.012, Figure 7B). During week ten of training the latency to approach the positive cue was significantly lower compared to the aversive cue (*Z* = −5.8, *p* = 0.000, Figure 8).

#### 3.1.2. Cue Location and Latency to Approach

An overall effect of cue location on latency to approach was observed in all pigs in both JBT1 and JBT2 (χ^2^(2) = 21.7, *p* = 0.000; Figure 9). During JBT1, an increased latency to approach was observed towards the aversive location compared to both the ambiguous (*Z* = −404.0, *p* = 0.000) and positive (*Z* = −3.88, *p* = 0.000) locations. Pigs further had increased latency towards the ambiguous location compared to the positive location (*Z* = −3.6, *p* = 0.020; Figure 9A). During JBT2, an increased latency to approach was observed towards the aversive location compared to both the ambiguous (*Z* = −3.99, *p* = 0.000) and positive (*Z* = −3. 7, *p* = 0.001) locations, but no increased latency towards the ambiguous location compared to the positive location was observed (*Z* = −1.4, *p* = 0.16; Figure 9B). Between JBT1 and JBT2, there was no difference in latency to approach the ambiguous location in pigs exposed to either enriched or barren housing treatments (*Z* = −1.2, *p* = 0.250 and *Z* = −1.22, *p* = 0.360 *p* = 0.36; Figure 9C). There was no significant effect of treatment on latency towards the ambiguous cue during JBT2 (*Z* = 2.11, *p* = 0.48; Figure 9D).

#### 3.1.3. Treatment Effects on Judgment Bias

No effect of treatment on JBI between JBT1 and JBT2 was observed (χ^2^ (20) = 2.0, *p* = 0.5).

#### 3.1.4. Blood and Brain MiRNA

At bleed 1 there were 51 differentially expressed miRNA between pigs exposed to enriched and barren housing (14 up regulated and 37 down regulated) but none were significant (FDR *p* > 0.05). Similarly, following bleed 2 there were 71 differentially expressed miRNA between pigs exposed to enriched and barren housing (43 up regulated and 28 down regulated) but none were significant at the FDR threshold (FDR *p* > 0.05). Within the amygdala, a total of 185 miRNA were differentially expressed (122 up regulated and 63 down regulated), but no significant effect of treatment was observed (FDR *p* > 0.05). The top 10 genes that were closest to achieving statistical significance, for each comparison, are listed in Appendix A.

#### 3.1.5. Dopamine, Serotonin and Metabolites

Pigs exposed to enriched housing had an increased concentration of dopamine (DA) (2838.8 ng/g vs. 1002.3 ng/g, *Z* = −2.26, *p* = 0.02) and its metabolite DOPAC (620.1 ng/g vs. 266.6 ng/g, *Z* = −2.26, *p* = 0.02) within the striatum, compared to pigs housed in barren conditions (Figure 10). No significant effect on DA or DOPAC was observed in the amygdala (Z = −0.94, *p* = 0.37 and Z = −0.53, *p* = 0.68) or prefrontal cortex (Z = −1.60, *p* = 0.37 and Z = −1.60, *p* = 0.37). Furthermore, treatment had no significant effect on serotonin (5HT) or its metabolite 5-HIAA in the striatum (Z = −0.8, *p* = 0.12), amygdala (Z = −1.60, *p* = 0.13), or prefrontal cortex (Z = −1.2, *p* = 0.68).

## 4. Discussion

In this study we investigated the suitability of circulating miRNA as biomarkers to distinguish valence of emotional state in the pig. We proposed that miRNA would be differentially expressed in the brain and blood during positive emotional states, and that a change in miRNA could be corroborated with already existing behavioural and physiological indices of emotional valence. We hypothesized that (i) exposing pigs to enriched housing conditions would result in a more positive judgment bias, increased neurotransmitter concentration, and differential miRNA patterns in the brain and blood compared to pigs exposed to barren environments, (ii) that changes in the expression of miRNA in the brain could be corroborated with changes of miRNA expression in blood, allowing peripheral miRNA response to be used as a proxy marker for emotional state in the pig. We found that treatment had no effect on behaviour during the JBT, nor did we observe differences in miRNA profiles in the brain or blood of pigs. There was an increase in concentrations of DA and its metabolite DOPAC in the striatum, but this increase was not observed in amygdala or prefrontal cortex. No difference in the neurotransmitter serotonin (5-hydroxytryptophan or 5-HT), nor its metabolite 5-HIAA, was found in any brain region between treatment groups. The results of this study imply that either (a) miRNAs are not likely to be valid biomarkers of positive affective state, at least under the type of conditions employed in this study, or (b) that the study design employed with enriched housing versus barren housing as a modifier of affective state was not sufficient to create differential affective states, and therefore establish the validity of miRNA as biomarkers.

With regard to the first possible interpretation—that miRNAs are not likely to be valid biomarkers of affective state—there is some limited evidence from the porcine literature on the validity of miRNAs, at least as biomarkers of negative states. Weaning stress [50], and heat stress [52], altered miRNA expression in intestinal and muscle tissue respectively. Lecchi et al. 2020 [53], also demonstrated that certain miRNA expression changes in saliva were present following castration and tail docking without analgesia. Our null finding, in contrast to these studies, might be explained by the assumed relatively low impact on physiological processes created in our study. The effects of heat and pain variously create cell damage, tissue degradation and inflammatory pathway activation which may not occur as a result of environmental change. MiRNA may therefore only be useful biomarkers where a relatively invasive change occurs that has a notable effect on physiology. 

A significant increase in the tonal concentration of DA, and its metabolite DOPAC, in the striatum of animals exposed to enriched housing conditions was observed. This finding is consistent with our hypothesis and suggests that the provision of enrichment resulted in a chronic shift in affective state, leading to a more positive emotional state in the animals. It is difficult to know if the relationship between the treatment and increased DA was a causative effect, or perhaps a response elicited by another biological process. Given that DA is implicated in behavioural control and is essential for reward related processes including reward learning [54,55], we anticipated this same difference to be reflected in the judgment bias data. For example, here we observed a treatment effect on tonal DA (i.e., a sustained level of DA neuron firing) where enriched housing increased tonal DA compared to animals housed in barren conditions. Subsequently, we would anticipate that the tonal increase in DA would influence behaviour, where pigs would, under ambiguity, have an enhanced expectation of a positive outcome and behave in a manner normally associated with a positive reward. Here, we did not detect a treatment effect on behavioural parameters; however, potential issues with the design of the behaviour paradigm may account for this and are discussed below. Furthermore, it is interesting that we did not see an increase in DA in the amygdala or the prefrontal cortex. Following rewarding experiences, dopaminergic neurons project widely throughout the brain. The ventral striatum is the region of the brain most closely associated with reward processing such as reward-based learning [56], and is directly innovated by the orbital prefrontal cortex and amygdala [57]. The amygdala plays a critical role in the coordination of the conscious experience of emotion and, along with the prefrontal cortex, forms reciprocal connections that allow learning and experience of the cognitive aspects of emotion [58]. It is unusual, then, given the interconnections between these regions, that no increase in DA was apparent in the amygdala or prefrontal cortex. However, there is some evidence from human studies that an increased reactivity in the ventral striatum occurs during adolescence, leading to stronger striatal activation in response to primary, secondary and social rewards [56]. We speculate that the age of the pigs used in the present study may have resulted in similar effects, where enhanced activity within the striatum may have occurred but was obscured in other brain regions (i.e., amygdala and prefrontal cortex) due to potential developmental differences in the brain. Further research is necessary to clarify and confirm this.

Serotonin is a key neurotransmitter abundant throughout the body and involved in a number of biological systems. Central 5-HT, however, is implicated in behavioural and neuropsychological processes including, but not limited to, mood regulation, appetite, sexuality and attention. In humans, chronic dysregulation of serotonergic activity, including alterations in serotonergic tone, is considered a key component underlying a number of affective disorders including anxiety and depression [59,60]. Serotonergic neurons originating from the raphe nucleus project to multiple brain structures involved in emotional regulation and behaviour response; this includes the amygdala [61], striatum [62], and prefrontal cortex [63]. Previously, administration of the 5-HT antagonist pCPA resulted in pessimistic judgment bias in sheep [44] and pigs [64], and depleted 5-HT concentration in brain regions including the rostral anterior cingulate cortex, prefrontal cortex, striatum, amygdala, hippocampus, hypothalamus and brain stem [65]. Furthermore, pharmacologically induced increases in 5-HT led to a positive judgment bias in rats with a dose dependent response [65]. Unexpectedly, we observed no difference in tonal 5-HT concentrations in the brain of pigs housed in enriched conditions. An explanation for this may be that the duration animals were exposed to the enriched treatment (four weeks), or the enrichment itself, was not sufficient to alter tonal 5-HT concentrations. Another factor may be that alterations in 5-HT levels are more closely associated with the body’s stress systems, including HPA activity in response to negative stimuli [66]. For example, following acute handling stress, 5-HT has been shown to be reduced from baseline levels in hippocampus and amygdala in fearful pigs, with the same reduction not occurring following non-stressful handling [66]. Another study has shown hippocampal 5-HT is positively correlated with standing alert time (freezing) during a novel object test, indicating a higher level of anxiety or fear in pigs [67]. It is plausible that the effect of enrichment was not sufficient to stimulate the bodies HPA axis, and thus no chronic changes in 5-HT levels were observed.

We expected that animals housed in enriched conditions would experience a more positive emotional state leading to the judgment of ambiguous stimuli with an enhanced expectation of a positive outcome, and, therefore, result in reduced time to approach the ambiguous cue provided. However, in this study no change in judgment bias was observed in response to enriched housing. There are two likely reasons for this: (i) there was no change in affective state in response to the treatments and/or (ii) the possibility that factors related to the training and test design may have compromised the JBT results.

Whilst increased space allowance, as provided in the enriched housing, has been shown to have beneficial effects on welfare in several studies [68], enrichment may be a determining factor in effects observed. Although the provision of enrichment has been previously shown to improve welfare outcomes and induce a positive bias in pigs [16,41,69], the type of enrichment given in this trial may not have been considered a rewarding stimulus by the pigs, and thus not been integrated at a cellular level. For example, for enrichment to be effective it should stimulate an animal’s visual, somatosensory, and olfactory systems whilst maintaining its novelty [70], where natural substrates, such as straw, green fodder, root vegetables and pressed or chopped miscanthus, are considered optimal for animal welfare. Unfortunately, the use of natural substrates for enrichment was not feasible in this trial due to the negative impact this may have had on the effluent system on this particular farm. Consequently, the substrate used may not have been sufficient to provide a rewarding stimulus. Furthermore, the provision of enrichment may have, in fact, affected the pigs in a negative manner, perhaps leading to aggression due to competition for the limited resource. Furthermore, the social structure of pigs is based on a dominance hierarchy, which is vigorously established through fighting when unacquainted pigs are brought together [71]. Although pen mates in the enriched housing group remained the same throughout this experiment, there may have been some incidences of aggression following training or testing, as individual animals were frequently removed from and then reintroduced to the group. Competition for resources could also have been a factor of disturbance for the pigs housed in groups. If the objects provided were insufficient then the social competition from pen mates may not allow all animals to use the enrichment at the same time, leading to adverse events such as aggression and tail biting [72]. It would have been beneficial to make additional behavioural observations of individuals in the enriched housing treatment to gain a better understanding of the level of activity and types of behaviour shown toward enrichment objects, as well as an account of behaviours considered to reflect positive emotions such as play behaviours [73,74].

Similar issues may have arisen in pigs housed in barren conditions. We would expect that that the effect of isolation in a barren environment would have a negative impact on the pigs and result in a more negative judgment bias. It may be that the animals exposed to barren environments did not find the environment extreme enough to alter behavioural outcomes in the judgment bias test. This has been observed in piglets where repeated social isolation had no effect on behaviour parameters toward ambiguous stimuli [75]. It may also be the case that the pigs housed in the negative environment were displaying rebound behaviour during the test. Rebound behaviour can be described as an increased tendency to perform a specific behaviour, i.e., an activity rebound, after a period of prevention [76]. If pigs were unable to perform locomotive behaviour due to the isolated and restricted housing, they may have developed or built up the urge to display increased locomotive behaviours once released into the test arena. If the pigs that were confined showed increased locomotive behaviour due to rebound effects, some may have touched the ambiguous probe (through choice or accidentally) quicker than if they were not confined, and thus confounded the latency to approach results. The test design itself may also have not been sensitive enough to successfully identify differences in affective state in the pigs in response to the housing treatment. During testing, a number of factors may have arisen which could have affected latency outcomes. It is common for judgment bias trials, including the present study, to leave the ambiguous cue unrewarded [9]. However, such an approach has, in some cases, led to loss of ambiguity towards the ambiguous cue and pigs learn to associate the ambiguous stimulus with an unrewarded outcome [9]. If pigs in this trial learned that the ambiguous stimulus was unrewarded during JBT1, and then remembered this during JBT2, their responses may have led to false measures of judgment bias, as seen previously in sheep [77] and pigs [78]. It has been suggested that rewarding ambiguous cues may maintain optimistic choices throughout testing [78], although similar issues may still arise through associative learning in relation to ambiguous trials that are rewarded. Furthermore, it has been suggested that the measurement of latency alone may lead to the false detection of pessimism in cases where animals are exposed to repeated ambiguous trials [79]. This was observed in rats, where exposure to repeated ambiguous trials was associated with increased latency. However, this increase in latency was also associated with optimistic responses in an active choice test [79]. As the authors in this study conclude, modification to the experimental designs that include both active-choice and latency measures would have been beneficial to minimize ambiguity of interpretation of latency data.

## 5. Conclusions

No changes in miRNA profiles in the brain or blood of pigs were observed in pigs exposed to either enriched or barren housing conditions. Although increased concentrations of dopamine and its metabolite DOPAC were observed in the striatum, this was not the case in the amygdala or prefrontal cortex. There was no difference is the neurotransmitter serotonin nor its metabolite 5-HIAA in any brain region between treatment groups. No difference was observed in judgment bias in any treatment group. There are two likely reasons for this: (i) there was no change in affective state in response to the treatments and/or (ii) the possibility that factors related to the training and test design may have compromised study outcomes. Therefore, in the absence of an adjunct measure indicative of valence of response (i.e., behavioural and physiological indices), we are unable to confirm the validity of miRNA as biomarkers of emotional state. However, given their promise as suggested in the literature, we recommend that further investigation of their utility as biomarkers for positive affective state should be undertaken.

## Figures and Tables

**Figure 1 animals-11-02054-f001:**
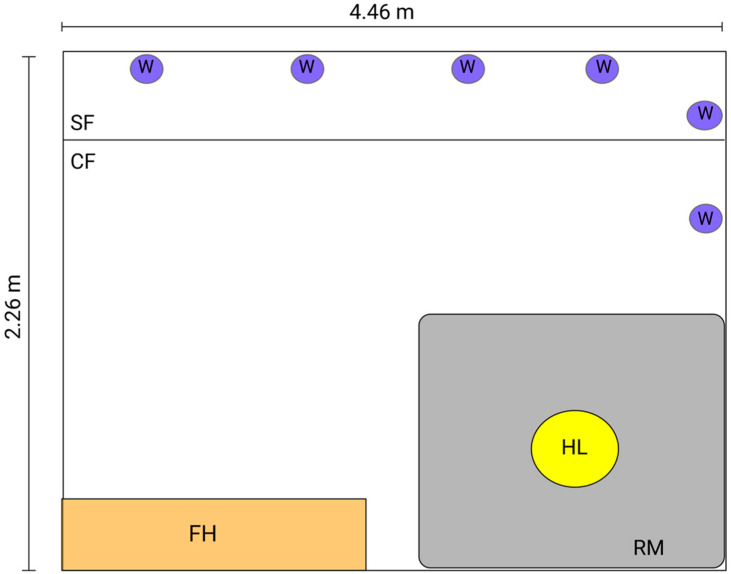
Group housing at Roseworthy piggery with six animals per pen. The pens were 2.26 m (W) × 4.46 m (L) × 0.86 m (H). The flooring consisted of half concrete (CF) and half slatted floor (SF). Each pen contained one feed hopper (FH), runner matting (RM), 6 nipple drinkers (W), and overhanging heat lamp (HL).

**Figure 2 animals-11-02054-f002:**
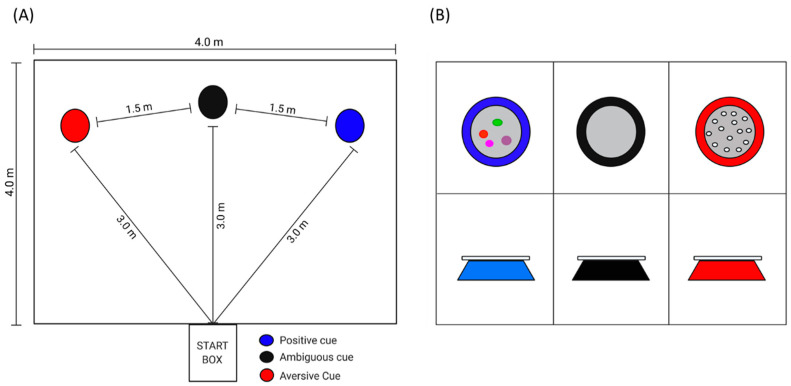
Illustrates (**A**) training arena for spatial go/no go task in pigs with positive, aversive and ambiguous cue locations depicted, (**B**) showing positive (blue bowl with food reinforcer), ambiguous (black bowl, no food reinforcer), and aversive (red bowl, no food reward plus a scare from human) stimulus.

**Figure 3 animals-11-02054-f003:**
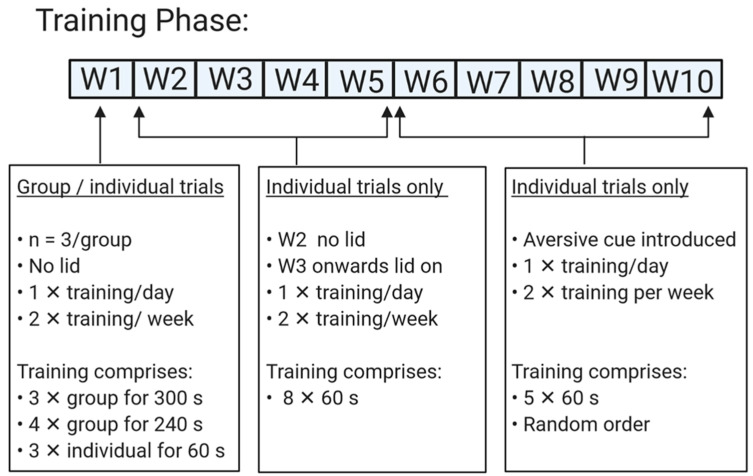
Indicates simple timeline of training protocol for spatial go/no go task where pigs were trained for a ten-week period in both group and individual trials.

**Figure 4 animals-11-02054-f004:**
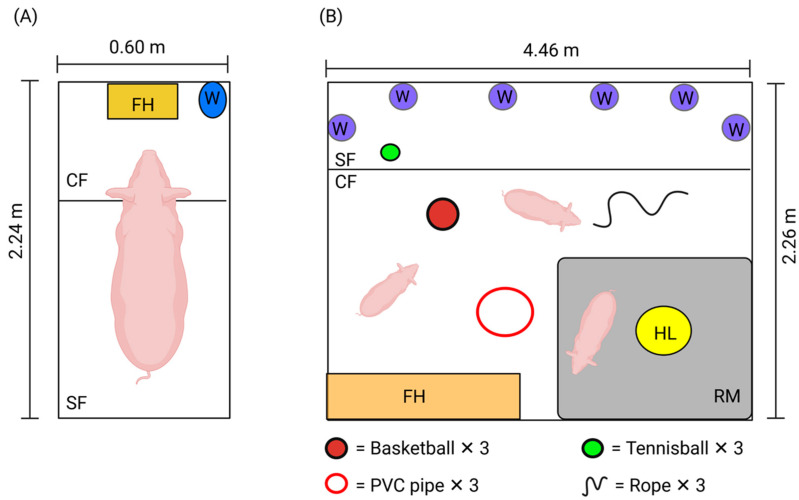
Indicates (**A**) barren housing (individually stalled, no human contact and no enrichment) and (**B**) enriched housing (group housed in pens, positive human interaction and enrichment provided). Barren housing conditions contained a feed hopper (FH), nipple drinker (W), concrete (CF), and slatted flooring (SF). Enriched housing contained a feed hopper (FH), nipple drinkers (W), concrete (CF), and slatted flooring (SF), a heat lamp (HL), rubber matting (RM) and enrichment materials (see legend in figure).

**Figure 5 animals-11-02054-f005:**
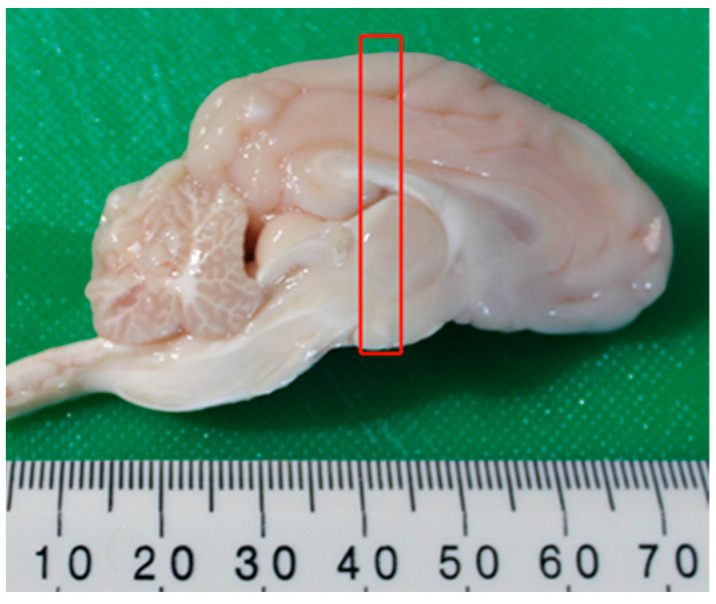
Example of the left cerebral hemisphere of the pig brains recovered in the experiments. The brain was further sliced into 5 mm coronal sections and placed into RNA stabilizing solution. The red box shows the approximate location of the 5 mm section where tissue from the amygdala was obtained.

**Figure 6 animals-11-02054-f006:**
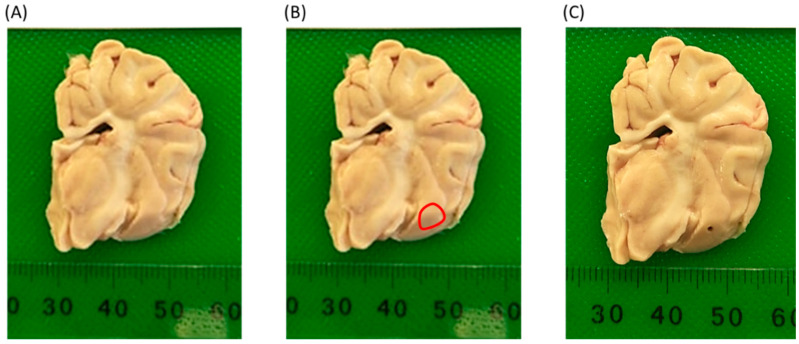
Sampling location for amygdala tissue, (**A**) Left hemisphere coronal section of pig brain with entire amygdala; (**B**) identification of amygdala (red circle), from pig atlas derived from Felix et al. [49]; (**C**) left hemisphere coronal section of pig brain with amygdala sample removed by punch biopsy.

**Figure 7 animals-11-02054-f007:**
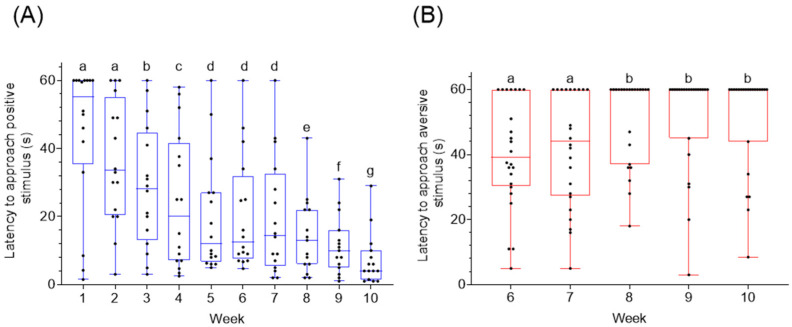
(**A**) Latency to approach (s) the positive stimulus during training weeks (1–10) in pigs (*n* = 24), (**B**) indicates latency to approach aversive stimulus during training weeks (6–10) in pigs (n24). Data are medians with range. Significant difference is indicated with differences in subscripts (*p* < 0.05).

**Figure 8 animals-11-02054-f008:**
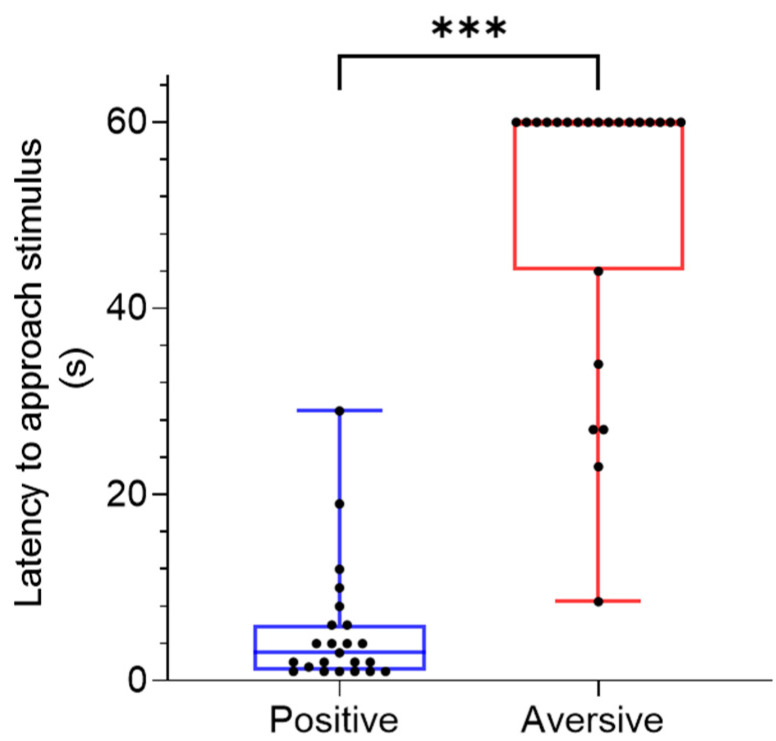
Latency to approach (s) positive and aversive stimulus at week ten of training in pigs (*n* = 23). Data are medians with range. Significant difference is indicated with presence of asterisk (*p* < 0.05).

**Figure 9 animals-11-02054-f009:**
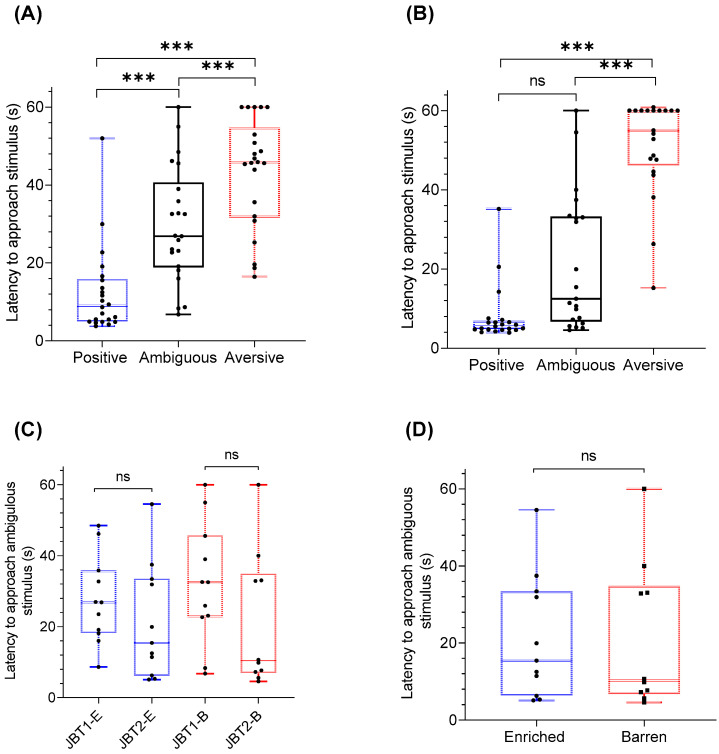
(**A**) Latency for pigs (*n* = 23) to approach positive, aversive and ambiguous stimulus at judgment bias test 1 (JBT1), (**B**) latency to approach positive, aversive and ambiguous stimulus in judgment bias test 2 (JBT2) in pigs exposed to enriched housing (*n* = 12), or barren housing (*n* = 11), (**C**) indicates latency to approach ambiguous stimulus between JBT1 and JBT2 in pigs exposed to enriched (*n* = 12), or barren housing (*n* = 11), and (**D**) indicates latency to approach ambiguous stimulus during JBT2 in pigs exposed to enriched (*n* = 12) or barren housing (*n* = 11). Data are medians with range. Significant difference is indicated with presence of asterisk (*p* < 0.05).

**Figure 10 animals-11-02054-f010:**
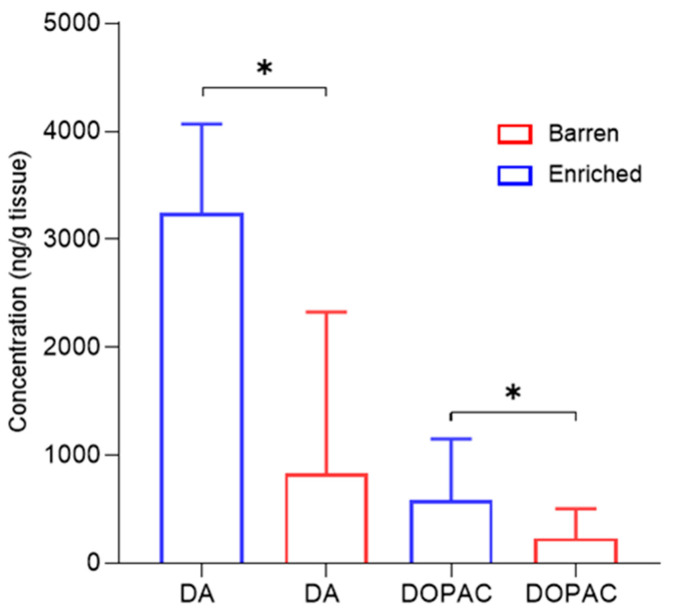
Concentration (ng/g tissue) of dopamine (DA), and its metabolite (DOPAC), in the striatum of pigs exposed to either enriched (*n* = 6) or barren (*n* = 6) housing treatments. Data are median ± range. Significant differences are indicated by presence of asterisks (*p* < 0.05).

## Data Availability

Data are available upon request from corresponding author.

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
