# Peer review of "Evaluation of miRNA as Biomarkers of Emotional Valence in Pigs"

_animals, 2021, doi:10.3390/ani11072054_

Round 1

Reviewer 1 Report

Authors deserve my recognition for expressing that the results do not support the initial hypotesis. In ane case, fron the scientifica pointo of view it is an intringuing contribution to highlight and invalid pathway.

Author Response

  1. Authors deserve my recognition for expressing that the results do not support the initial hypotesis. In ane case, fron the scientifica pointo of view it is an intringuing contribution to highlight and invalid pathway.

Author responses:

  1. Thank-you reviewer 1 for your comment. The authors agree that despite the null finding in miRNA differentiation, the findings of this study provide an intriguing contribution to the area of animal welfare assessment, in particular the assessment of positive emotional state in the pig.

Reviewer 2 Report

General comments

I thank the authors for submitting a well crafted and thoughtful manuscript. There are little if any corrections to be made, beyond some minor ones noted below. I suspect the most difficult element is the “double negative” outcome here. Whilst there is evidence of the role of miRNA in responding to negative stimuli, the effect of positive stimuli is unclear (this is well discussed by the authors) so it is maybe of little surprise to see a lack of a positive outcome in this area. It is more unusual (and maybe unlucky?) to see a lack of treatment effect. Recent work such as that of Hayes et al * would suggest that the treatment here would have been sufficient to provoke a significant outcome, which would have clearly made this an easier manuscript. However, I am aware that a non-significant finding is as important in some instances as a significant one, and this paper does add to the search for a suitable physiological measure of affective states.

Minor comments

Abstract – would be good to see the significance of the dopamine findings (p value) in the abstract in my opinion.

L22, 28 etc. Behaviour vs behavior. Suggest a “find and replace” through the paper to make sure that behaviour is spelt consistently correctly.

L23 – makes comment as to other methods being “unsuited to production environments” which does beg the question is this method any more so. Not a change per se more an interest.

L163 – “JB paradigm”. I know that strictly speaking the tests themselves are really the true definition of “JBT” but I wonder if paradigm is the right word here vs “test”, for consistency.

L195 – “chocolate treats”. I think “sweet treats” are used in L170. Consistency, especially for readers of either a non-English language background (or even Americans). Sweets vs lollies vs candies vs chocolate might all be used to describe M&Ms in different places. Chocolate treats is maybe clearest?

L420 – JBI instead of JBT? Unless my eyesight is going.

*Effects of Positive Human Contact during Gestation on the Behaviour, Physiology and Reproductive Performance of Sows by Megan E. Hayes, Lauren M. Hemsworth, Rebecca S. Morrison, Kym L. Butler, Maxine Rice, Jean-Loup Rault and Paul H. Hemsworth Animals 2021, 11(1), 214

Author Response

comments:

  1. Abstract – would be good to see the significance of the dopamine findings (p value) in the abstract in my opinion.
  2. L22, 28 etc. Behaviour vs behavior. Suggest a “find and replace” through the paper to make sure that behaviour is spelt consistently correctly.
  3. L23 – makes comment as to other methods being “unsuited to production environments” which does beg the question is this method any more so. Not a change per se more an interest.
  4. L163 – “JB paradigm”. I know that strictly speaking the tests themselves are really the true definition of “JBT” but I wonder if paradigm is the right word here vs “test”, for consistency.
  5. L195 – “chocolate treats”. I think “sweet treats” are used in L170. Consistency, especially for readers of either a non-English language background (or even Americans). Sweets vs lollies vs candies vs chocolate might all be used to describe M&Ms in different places. Chocolate treats is maybe clearest?
  6. L420 – JBI instead of JBT? Unless my eyesight is going.

Author Response:

Thank-you reviewer 2 for your comments on our manuscript. Please see below responses to your comments listed above.

  1. Amended and highlighted in the abstract.
  2. Amended and highlighted throughout manuscript.
  3. Thank-you and yes that is a very good point. Often physiological markers can tell us about the emotional arousal of animals but cannot differ in valence of affect. Behavioural measures have been used to distinguish valence of affect, but require lengthy training times and thus not suited to production environments. Here we aim to investigate if the measure of miRNA expression in the blood can be used as a proxy marker for emotional state in the pig. Although collection of blood from individual animals (and subsequent miRNA analysis), may not be feasible to implement in a whole-farm context, validation of these biomarkers will potentially lead to the development of tools/technologies that can be used to assess the emotions of animals both individually and in a whole-group setting (i.e., various surveillance and sensory based technologies, i.e., in blood or saliva). Clarification in manuscript (see lines 92-96)
  4. Amended and highlighted in the manuscript.
  5. Amended and highlighted in the manuscript.
  6. JBI= Judgement bias index,

JBT- Judgement bias test (1 or 2)

Following the formula described by Horback et al [44], an individual judgment bias index (JBI), is calculated for each animal and is based on their responses in from JBT1 and JBT2 data. This allows possible intrinsic differences between pigs to be accounted for in the analysis.

Reviewer 3 Report

 In method,

-in line of 168 how do you make scare to pigs from human? I think you need give more explanation on how do you do it.

-for your refreshing training, you mentioned that "This was performed to ensure pigs remained familiar with testing arena between the first and second tests", can you explain that this procedure would not lead to the effect of the 1st trail' experience on the results of the 2nd trail?

-I think you need give the reason on why you use same groups of pigs for the two consecutive trials (W2-W5 vs W6-W10)? do you considerate the effect of experience (or expectation) on the latency to the bowls due to the same test room used ?

Author Response

Comments:

  1. in line of 168 how do you make scare to pigs from human? I think you need give more explanation on how do you do it.
  2. for your refreshing training, you mentioned that "This was performed to ensure pigs remained familiar with testing arena between the first and second tests", can you explain that this procedure would not lead to the effect of the 1st trail' experience on the results of the 2nd trail?
  3. I think you need give the reason on why you use same groups of pigs for the two consecutive trials (W2-W5 vs W6-W10)? do you considerate the effect of experience (or expectation) on the latency to the bowls due to the same test room used?

Author responses:

Thank-you reviewer 3 for your comments and consideration regarding our manuscript. Please see below responses in relation to your comments listed above.

  1. Please see lines 220-222 for details on human scare.
  2. During refresher training pigs were not exposed to the ambiguous stimulus and therefore should not have impacted on behaviour in the JBT2. Refresher training is designed to reinforce the associations between positive and aversive stimuli only, thus would not impact the behavioural responses from pigs when exposed to ambiguity. Authors have clarified this in the text (see lines 232-233). There may have been an effect of repeated testing on the behavioural response to the ambiguous stimulus between JBT1 and JBT2, and this was highlighted in the discussion as a potential limiting factor (see lines 597-614)
  3. All trials between W1-5 and W6-10 are training trials. This process of training is to ensure that all pigs are making the associations between positive and aversive stimuli. W1-5 the pigs learnt the association with the positive cues and W6-10 the pigs learn the association towards the aversive cues. Therefore, the same pigs are used throughout the training phase as it takes some time for the animals to learn these associations. We therefore expect the latency to approach would change over time as the animals learnt the associations (i.e., faster towards positive stimulus and slower towards negative stimulus).